# POINT BRIDGE: 3D REPRESENTATIONS FOR CROSS DOMAIN POLICY LEARNING

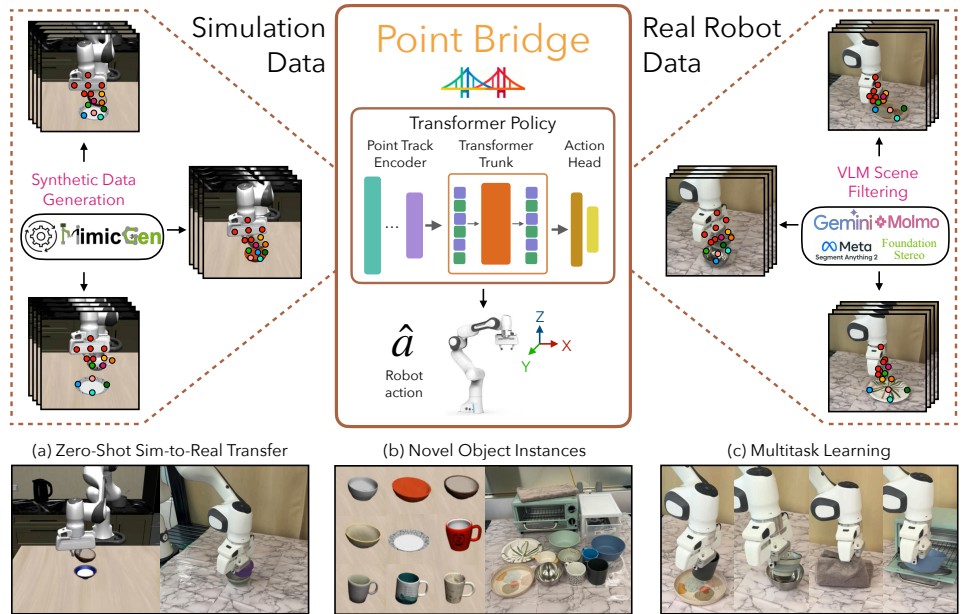

Figure 1: **POINT BRIDGE Overview.** We present POINT BRIDGE, a framework that leverages unified, domain-agnostic point-based representations to unlock the potential of large-scale synthetic simulation datasets. POINT BRIDGE enables zero-shot sim-to-real policy transfer with minimal visual or object alignment, supports multitask learning, and further improves performance when co-trained with small amounts of real robot data.

## ABSTRACT

Robot foundation models are starting to realize some of the promise of developing generalist robotic agents, but progress remains bottlenecked by the availability of large-scale real-world robotic manipulation datasets. Simulation and synthetic data generation are a promising alternative to address the need for data, but the utility of synthetic data for training visuomotor policies still remains limited due to the visual domain gap between the two domains. In this work, we introduce POINT BRIDGE, a framework that uses unified domain-agnostic point-based representations to unlock the potential of synthetic simulation datasets and enable zero-shot sim-to-real policy transfer without explicit visual or object-level alignment across domains. POINT BRIDGE combines automated point-based representation extraction via Vision-Language Models (VLMs), transformer-based policy learning, and inference-time pipelines that balance accuracy and computational efficiency to establish a system that can train capable real-world manipulation agents with purely synthetic data. POINT BRIDGE can further benefit from co-training on small sets of real-world demonstrations, training high-quality manipulation agents that substantially outperform prior vision-based sim-and-real co-training approaches. POINT BRIDGE yields improvements of up to 44% on zero-shot sim-to-real transfer and up to 66% when co-trained with a small amount of real data. POINT BRIDGE also facilitates multi-task learning. Videos of the robot are best viewed at: https://pointbridge-anon.github.io/

# 1 INTRODUCTION

Deep learning has recently undergone a paradigm shift, moving from narrow task-specific models to generalist systems capable of complex reasoning (Achiam et al., 2023; Team et al., 2023; Touvron et al., 2023), generating photorealistic images (Blattmann et al., 2023) and videos (Liu et al., 2024), and even writing code (Li et al., 2022). This progress has been fueled by internet-scale training data paired with scalable architectures. Lately, robot foundation models are starting to realize some of the promise of large-scale data and the training paradigm from these domains. However, unlike vision and language, which can directly exploit internet-scale datasets, robotics is inherently interactive: models must learn from datasets that contain physical interactions with the real world. This makes collecting large-scale robotic data time-consuming, prohibitively expensive, and fundamentally difficult to scale, creating a central bottleneck for building generalist robotic intelligence.

The prevailing paradigm for robot policy learning relies on large-scale teleoperated datasets, followed by training neural policies on them. While effective, this approach often requires months or years of data collection and still produces datasets far smaller than those in vision and language (Goldberg, 2025). Simulation is a promising alternative to address this need for data, especially due to recent progress. Simulation environments are becoming easier to design, with the availability of high-fidelity physics simulators (Todorov et al., 2012; Mittal et al., 2023) and the emergence of generative AI tools that automate asset and scene generation (Wang et al., 2023; Nasiriany et al., 2024). Recently developed synthetic data generation tools can generate large-scale, high-quality robot manipulation demonstration datasets in such simulation environments with minimal human effort (Dalal et al., 2023; Mandlekar et al., 2023; Jiang et al., 2024; Garrett et al., 2024). Furthermore, recent work has shown that such synthetic simulation datasets can easily train high-performing real-world manipulation agents by co-training on these datasets and small numbers of real-world demonstrations (Maddukuri et al., 2025; Wei et al., 2025; Bjorck et al., 2025), suggesting that synthetic simulation data could potentially reduce the dependence on large real-world datasets. However, these methods can still require careful sim and real alignment, and still rely on the presence of real-world data, owing to the mismatched representation of data between the domains. Human videos offer another scalable and complementary source of supervision, but again face challenges from the embodiment gap between human and robot morphologies as well as the representation mismatch between the domains.

A recent line of work proposes task-relevant keypoint representations (Haldar & Pinto, 2025; Zhu et al., 2024; Liu et al., 2025) as a potential solution to this domain representation gap. By abstracting both the robot and scene into sets of keypoints, these methods enable policies that are agnostic to raw visual appearance and generalize across objects and environment conditions. However, existing approaches often rely on human annotations (Haldar & Pinto, 2025; Liu et al., 2025), focus on bridging embodiment but not visual differences (Lepert et al., 2025b;a), and are often restricted to single-task settings. We argue that such representations only scratch the surface of what is possible.

**In this work, we introduce POINT BRIDGE, a framework that uses unified domain-agnostic point-based representations to unlock the potential of synthetic simulation datasets and enable zero-shot sim-to-real policy transfer.** POINT BRIDGE trains real-world manipulation agents starting with just a handful of teleoperated demonstrations in simulation by using synthetic data generation tools. It then leverages advances in vision-language models (VLMs) to build unified scene representations that facilitate cross-domain policy transfer. Our core insight is that unifying representations across simulation and real-robot teleoperation unlocks scalable sim-to-real transfer without requiring explicit visual or object-level alignment. Such a representation further supports scaling to multi-task policies through transformer-based architectures, providing a framework that scales with data availability. POINT BRIDGE operates in three stages. First, scenes are filtered into point cloud–based representations aligned to a common reference frame. In simulation, this is obtained directly from object meshes, while in real experiments, we use our automated VLM-guided pipeline for keypoint extraction on task relevant objects. Second, a transformer-based policy architecture is trained on these unified point clouds for policy learning. Finally, during deployment, we employ a lightweight pipeline for scene extraction designed to minimize the sim-to-real gap, leveraging VLM filtering and supporting multiple 3D sensing strategies to balance performance and throughput.

We demonstrate the effectiveness of POINT BRIDGE on six real-world tasks, using data collected through simulation and real robot teleoperation. Our main findings are as follows:

1. We develop POINT BRIDGE, a framework that uses unified domain-agnostic point-based representations to harness synthetic simulation data and enable zero-shot sim-to-real policy transfer.

2. POINT BRIDGE contains novel components including (1) a VLM-based point extraction pipeline that bridges the visual sim-to-real gap with minimal human effort, and (2) multiple inference-time pipelines to adapt to different user needs with respect to performance and throughout.

3. POINT BRIDGE improves by 39% and 44% on single-task and multitask zero-shot sim-to-real transfer. When co-trained with a small amount of real data, POINT BRIDGE improves over prior works by 61% and 66% in single-task and multitask settings, respectively. (Section 5.2, 5.3).

4. We present a systematic analysis of key design choices in POINT BRIDGE (Section 5.4).

All of our datasets, training, and evaluation code will be made publicly available. Videos of our trained policies are best viewed at: https://pointbridge-anon.github.io/.

## 2 RELATED WORK

### 2.1 STRUCTURED REPRESENTATIONS

Structured representations of scene elements enable more efficient and semantically meaningful learning. Common techniques include segmentation into bounding boxes (Devin et al., 2018; Zhu et al., 2023b) and object pose estimation (Tremblay et al., 2018; Tyree et al., 2022). Bounding boxes show promise but suffer from overfitting to specific instances, while pose estimation is less prone to this but requires separate models per object. Point clouds (Zhu et al., 2023a; Bauer et al., 2021) are a popular alternative but their unstructured nature complicates learning spatial relationships. Recently, key points (Levy et al., 2025; Ju et al., 2024; Huang et al., 2024; Haldar & Pinto, 2025; Fang et al., 2025; Ren et al., 2025) have gained traction for policy learning due to their generalizability and support for direct human prior injection (Bharadhwaj et al., 2024b;a), contrasting with approaches that first learn representations from human videos followed by robot teleoperation data (Nair et al., 2022; Wu et al., 2023; Ma et al., 2022; 2023; Karamcheti et al., 2023).

### 2.2 DATA COLLECTION AND GENERATION FOR ROBOTICS

Robot teleoperation (Mandlekar et al., 2018; Wu et al., 2024; Zhao et al., 2023b; Iyer et al., 2024) is a popular method for collecting task demonstrations – here, humans use a teleoperation device to control a robot and guide it through tasks. Several efforts (Brohan et al., 2022; Ebert et al., 2021; Brohan et al., 2023) have scaled up this paradigm by using a large number of human operators and robot arms over extended periods of time (e.g., months). Some works have also allowed for robot-free data collection with specialized hardware (Chi et al., 2024; Fang et al., 2023; Shafiullah et al., 2023), but human effort is still required for data collection. Other works seek to generate datasets automatically using pre-programmed demonstrators in simulation (Dalal et al., 2023; James et al., 2020; Ha et al., 2023), but scaling these approaches to a larger variety of tasks can be difficult.

### 2.3 LEARNING MANIPULATION FROM HUMAN DEMONSTRATIONS

Behavioral Cloning (BC) (Pomerleau, 1988; Ross et al., 2011) is a method for learning policies offline from demonstrations using supervised learning. Recent advances in BC have demonstrated success in learning policies for both long-horizon tasks (Mandlekar et al., 2021; 2020; Shridhar et al., 2021) and multi-task scenarios (Haldar et al., 2024; Bharadhwaj et al., 2023a; Padalkar et al., 2023; Bharadhwaj et al., 2024b;a). However, most of these approaches rely on image-based representations (Zhang et al., 2018; Haldar et al., 2024; Chi et al., 2023; Bharadhwaj et al., 2023b; Padalkar et al., 2023), which limits their ability to generalize to new objects and function effectively outside of controlled lab environments. A way to make policies generalize better is to leverage offline data augmentation to increase the size of the training dataset for learning policies (Zhan et al., 2021; Yu et al., 2023; Chen et al., 2023; Bharadhwaj et al., 2023a; Zhao et al., 2025).

## 2.4 SIM-TO-REAL POLICY TRANSFER

Sim-to-real policy transfer aims to enable models trained in simulation to perform well in the real world. A common method is domain randomization (Zhu et al., 2018; Andrychowicz et al., 2020; Handa et al., 2023), which introduces variability in simulation to train policies robust to simulation-reality gaps. However, it often requires careful tuning and substantial human effort to define effective randomization ranges. Another approach minimizes this gap by enhancing simulation fidelity via system identification (Ramos et al., 2019; Muratore et al., 2022; Lim et al., 2022; Memmel et al., 2024; Kumar et al., 2021; Evans et al., 2022) and digital twins (Jiang et al., 2022; Torne et al., 2024), aligning simulation with real dynamics. These methods also demand significant manual effort, limiting their applicability across diverse tasks. Recent work trains real-world manipulation policies using mixed simulation and real data (Bjorck et al., 2025; Nasiriany et al., 2024; Zitkovich et al., 2023; Ankile et al., 2024), outperforming policies trained on real data alone. Moreover, simulation data need not perfectly match reality, making this a compelling alternative.

## 3 PREREQUISITES

**Learning from Demonstrations.** The goal of imitation learning is to learn a behavior policy $\pi : \mathcal{O} \to \mathcal{A}$ from a dataset of $N$ expert demonstrations, denoted as $\mathcal{T}^e = \{(o_t, a_t)_{t=0}^T\}_{n=1}^N$, where $o_t \in \mathcal{O}$ and $a_t \in \mathcal{A}$ represent the observation and action at timestep $t$, and $T$ is the horizon length of each episode. The behavior policy is trained using Behavior Cloning (Pomerleau, 1988) by maximizing the log-likelihood of expert actions, i.e.,

$$\theta^* = \arg\max_\theta \sum_{n=1}^N \sum_{t=0}^T \log \pi_\theta(a_t^n \mid o_t^n),$$

where $\pi_\theta$ is the parameterized policy and $\theta$ are the learnable parameters.

**Problem Statement** Our goal is to leverage a source dataset $\mathcal{D}_{src} = \{\tau_{src}^i\}_{i=1}^N$ of human demonstrations for a task $\mathcal{M}$ in simulation, where each trajectory $\tau_{src}^i = \{(o_t, a_t)\}_{t=0}^T$ consists of observations $o_t \in \mathcal{O}$ and expert actions $a_t \in \mathcal{A}$. Using synthetic data generation techniques (Mandlekar et al., 2023), we expand $\mathcal{D}_{src}$ into a larger dataset $\mathcal{D}_{sim}$. The objective is to learn policies $\pi_\theta : \mathcal{O} \to \mathcal{A}$ on this data that can be deployed zero-shot in the real world. We also consider the case where a small set of real-world demonstrations $\mathcal{D}_{real}$ is available, enabling policies to be jointly trained on both simulated and real data to improve transfer. Finally, we explore the multitask setting, where a single policy is trained across multiple tasks $\{\mathcal{M}_1, \ldots, \mathcal{M}_K\}$ conditioned on task-specific instructions.

**Simulation Assumptions** For synthetic data generation in simulation, we make the following assumptions: (1) The dataset includes policy actions $\mathcal{A}$ consisting of continuous end-effector pose commands and a discrete gripper command. This allows each demonstration to be treated as a sequence of target poses for a task-space controller. (2) Each task involves a set of manipulable objects $\{O_1, \ldots, O_k\}$. (3) During data collection, the pose of each object can be observed or estimated before the robot makes contact.

**Real-World Assumptions** In real-world experiments, we assume a calibrated scene with known camera intrinsics and extrinsics. All 3D observations are expressed in a consistent reference frame, aligned with the robot arm's base frame at every timestep.

## 4 POINT BRIDGE

POINT BRIDGE introduces a unified scene representation that enables sim-to-real policy transfer with minimal alignment, incorporates co-training with real-world data, and facilitates multitask learning. An overview of the framework is provided below, with details discussed in the following sections.

## 4.1 OVERVIEW

POINT BRIDGE begins with a small dataset of human demonstrations $\mathcal{D}_{src}$, which is expanded into a larger dataset $\mathcal{D}_{sim}$ using synthetic data generation (Mandlekar et al., 2023). We also consider an

Figure 2: **Point Extraction Pipeline Overview.** Given a scene image and task description, Gemini (Team et al., 2023) identifies the task-relevant objects, which are then localized using Molmo (Deitke et al., 2024) and SAM-2 (Ravi et al., 2024) Subsequently, 3D keypoints on these objects are generated by uniformly sampling 2D keypoints on the image and projecting them into 3D using depth from Foundation Stereo (Wen et al., 2025), together with camera intrinsics and extrinsics.

optional setting where a small set of real-world demonstrations $\mathcal{D}_{real}$ is available for co-training. All observations are converted into a compact point-based representation $\mathcal{P}$, serving as input to policies mapping observations to actions. In simulation, these representations are obtained directly from the simulator, while in the real world, they are extracted via a VLM-guided scene filtering pipeline. During deployment, the same VLM pipeline provides task-relevant points in real time for policy inference. The resulting policies enable zero-shot sim-to-real transfer, joint training with real data, and multitask learning. Details about each component are provided in the subsequent sections.

## 4.2    DATA COLLECTION AND SYNTHETIC DATA GENERATION

For our simulated tasks, we use the MimicLabs suite (Saxena et al., 2025) to construct atomic tasks, each involving different pairs of object instances. For each task, we collect a small set of human demonstrations $\mathcal{D}_{src}$, which are then expanded into a much larger dataset $\mathcal{D}_{sim}$ using MimicGen (Mandlekar et al., 2023), a synthetic data generation technique. MimicGen adapts each demonstration segment to novel scenes by applying a constant SE(3) transformation $T_W^{o_i'}(T_W^{o_i})^{-1}$, where $T_W^{o_i}$ is the pose of the source object $o_i$ in the world frame, and $T_W^{o_i'}$ is the pose of the same object in the target scene. The inverse transformation $(T_W^{o_i})^{-1}$ maps from the world frame to the source object's local frame, and the full product maps poses from the source object's frame to the target object's frame in the new scene. This transformation preserves the relative geometry between the end effector and the object from the source demonstration when adapting to new object poses. As a result, MimicGen enables a small set of demonstrations to be multiplied many times over with novel object configurations and types, supporting generalizable policy learning on large-scale datasets.

## 4.3    POINT EXTRACTION

Each observation in the dataset is now distilled into a compact set of task-relevant 3D keypoints. These keypoints serve as the unified representation used for downstream policy learning. The pipeline comprises two stages: (1) identifying task-relevant objects in the scene, and (2) extracting 3D keypoints for those objects. An overview of this pipeline is shown in Figure 2.

**VLM-Guided Scene Filtering**    Given an initial scene image $\mathcal{I}_0$ and a natural language task description $\mathcal{L}$, we first use `Gemini-2.5-flash` to identify the set of task-relevant objects in the scene, denoted as $\{l^1, \ldots, l^k\}$. For example, for the command *"put the bowl on the plate"*, the model returns the object set bowl, plate. After determining the object categories, we employ Molmo-7B (Deitke et al., 2024) to localize these objects as pixels $\{o^{p_1}, \ldots, o^{p_k}\}$ in the image.[1] These pixel coordinates serve as initialization for SAM2 (Ravi et al., 2024), which extracts 2D segmentation masks $\{m_0^1, \ldots, m_0^k\}$ for each identified object. For subsequent frames in the trajectory, we leverage SAM2's built-in memory to propagate masks consistently and track objects robustly over time, enabling reliable handling of occlusions during both data collection and deployment.

---

[1]In our experiments, `Gemini-2.5-flash` was effective for text-based object identification but less reliable for spatial localization, motivating the use of a specialized VLM for the pointing task. As multi-modal VLMs advance, a unified model could eventually replace this modular approach.

**3D Projection of Task Objects**    For each timestep $t$, $N$ 2D object points $\mathcal{P}_t^{2D}$ are sampled uniformly from each object segmentation mask $m_t^i, \forall i \in \{1, \ldots, k\}$. A stereo image pair of the scene is then used to compute depth $\mathcal{I}_t^d$ with Foundation Stereo (Wen et al., 2025). This depth map, along with camera intrinsics and extrinsics, lifts $\mathcal{P}_t^{2D}$ to 3D. FoundationStereo generally produces less noisy depth than commodity RGB-D sensors, especially for shiny or transparent objects. To reduce redundancy while maintaining coverage, we apply farthest point sampling to downsample each object to $M$ ($\ll N$) representative points. Finally, all object points are transformed into the robot base frame using camera extrinsics. We denote the final set as $\mathcal{P}_t^{3D}$.

**Considerations for simulation data**    In simulation, we bypass VLM-based detection and directly sample 3D points from task-relevant object meshes. However, mesh-based sampling covers all object surfaces, while real cameras only capture visible surfaces from specific viewpoints. To bridge this gap, we replicate real camera setups by applying the corresponding extrinsic $(R, t)$ and intrinsic $(K)$ parameters. Each mesh point $X_\text{mesh}$ is projected to the image plane as $\tilde{x} = K[R|t]X_\text{mesh}$, and the pixel coordinate is $x = (\tilde{x}_1/\tilde{x}_3, \ \tilde{x}_2/\tilde{x}_3)$. We then use the ground-truth depth map $D(x)$ to lift the point back to 3D: $X_\text{cam} = D(x)K^{-1}[x\ 1]$. These points are transformed into the robot's base frame for consistency. Finally, to account for sensor noise absent in simulation, we inject Gaussian noise with a $1$ cm standard deviation into the point clouds to improve robustness to real-world observations.

**Robot Representation**    Similar to Haldar & Pinto (2025), we represent the robot end effector as a set of keypoints on the gripper. Given the robot pose $T_r^t$ at timestep $t$, we define $N$ rigid transformations $T$ about this pose and compute the pose at each robot keypoint $T_r^t$ such that

$$(T_r^t)^i = T_r^t \cdot T^i, \ \ \forall i \in \{1, ..., N\} \tag{1}$$

The positions of the robot key points $(\mathcal{P}_r^t)^i \ \ \forall i \in \{1, ..., N\}$ are then extracted from these poses.

## 4.4 Policy Learning

We use BAKU (Haldar et al., 2024) for policy learning . Robot points $\mathcal{P}_r$ and object points $\mathcal{P}_o$ are combined into a point cloud $\mathcal{P}$, encoded with a PointNet (Qi et al., 2017) encoder. For multitask learning, we also input a language embedding $\mathcal{L}$, encoded using the 6-layer MiniLM (Wang et al., 2020) from Sentence Transformers (Reimers & Gurevych, 2019). The encoded representations serve as input to a BAKU transformer policy with a deterministic action head that outputs the robot end-effector pose and gripper state. Mathematically,

$$\begin{aligned} \mathcal{O}^{t-H:t} &= \{\mathcal{P}_r^{t-H:t}, \ \mathcal{P}_o^{t-H:t}, \ \mathcal{L}\} \\ \hat{\mathcal{A}}^{t+1} &= \pi(\cdot \mid \mathcal{O}^{t-H:t}) \end{aligned} \tag{2}$$

where $H$ is the history length, $\pi$ the learned policy, and $\mathcal{A}$ the predicted action. Following prior work in policy learning (Zhao et al., 2023a; Chi et al., 2023), we use action chunking with exponential temporal averaging to ensure smoothness of the predicted tracks. The policy is optimized with mean squared error (MSE) over ground-truth and predicted actions.

## 4.5 Policy Inference

During real-world deployment, the initial scene image $\mathcal{I}_0$ and task instruction $\mathcal{L}$ are used to obtain 2D object keypoints $\mathcal{P}_0^{2D}$, which are projected to 3D using scene depth and camera parameters. Section 4.3 describes our primary approach with stereo images and Foundation Stereo, but we also support depth from commodity RGB-D sensors and point triangulation from two RGB cameras (Haldar & Pinto, 2025). For RGB-D sensors, depth comes directly from the sensor depth map. For triangulation, 2D keypoints from one camera view are transferred to the other via MAST3R (Leroy et al., 2024), and Co-Tracker (Karaev et al., 2023) tracks them throughout the trajectory. 3D keypoints $\mathcal{P}_0^{3D}$ are then computed by triangulating tracked corresponding 2D points from multiple views and transforming them into the robot base frame. In subsequent timesteps, Co-Tracker separately tracks 2D keypoints $\mathcal{P}_t^{2D}$ in both views, followed by multi-view triangulation to extract $\mathcal{P}_t^{3D}$ in the base frame. This flexible pipeline with multiple depth sensing strategies enables the same trained policy to be deployed across diverse real setups. We compare performance across strategies in Section 5.4.

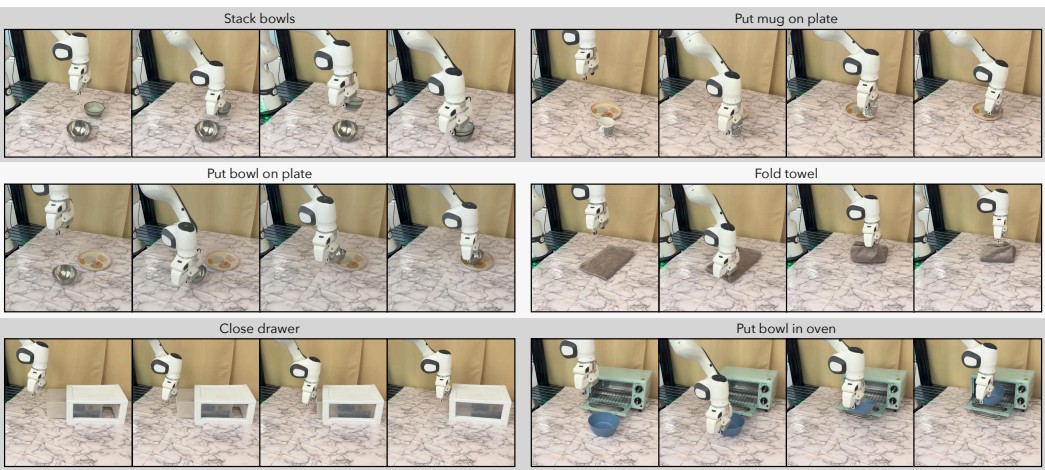

Figure 3: **Tasks.** Real-world rollouts showing POINT BRIDGE's ability on 6 real-world tasks.

## 5 EXPERIMENTS

We provide details on our experimental setup (Sec. 5.1) and subsequently show how POINT BRIDGE effectively enables zero-shot sim-to-real policy transfer from synthetic simulation data (Sec. 5.2) and how POINT BRIDGE performance can be improved even further with a small amount of real-world data (Sec. 5.3). Finally, we conduct a systematic analysis of the components in POINT BRIDGE (Sec. 5.4). We have included additional experiments and analysis in Appendix A.2.

### 5.1 EXPERIMENTAL SETUP

We evaluate manipulation tasks with significant variability in object type and placement, under minimal visual and object alignment between simulation and reality. We use Deoxys (Zhu et al., 2022) at 20 Hz as the robot controller. Real-world experiments are conducted on a Franka Research 3 arm with a Franka Hand gripper. Demonstrations are collected at 20 Hz using RoboTurk (Mandlekar et al., 2018) in simulation and Open Teach (Iyer et al., 2024) in the real world, and subsampled to 10 Hz for training. For sensing, we use an Intel RealSense RGB-D and a ZED 2i stereo camera. Policies trained with POINT BRIDGE and FoundationStereo for depth estimation run at 5 Hz, while image-based baselines reach 15 Hz. **In total, we perform 1410 real-world evaluations across varied task settings to benchmark performance.**

**Environment Design and Data Generation**    For our simulated experiments, we use the MimicLabs task suite (Saxena et al., 2025) to design 3 atomic tasks - `bowl on plate`, `mug on plate`, and `stack bowls`. Each task includes 4 different object instance pairs. For every pair, a human demonstrator provides 5 demonstrations, which are scaled up to 300 using MimicGen (Mandlekar et al., 2023), resulting in a total of 1200 demonstrations per task in simulation. For co-training, we supplement this with 45 teleoperated demonstrations in the real world across three additional object pairs, illustrating cross-domain variability. For real tasks such as `fold towel`, `close drawer`, and `put bowl in oven`, we only collect real-world data (20 demonstrations on a real robot). We provide additional details about policy learning considerations and task descriptions in Appendix A.2.

### 5.2 ZERO-SHOT SIM-TO-REAL TRANSFER WITH MINIMAL ALIGNMENT

We evaluate POINT BRIDGE for zero-shot sim-to-real transfer on 3 simulated tasks. Table 1 and Table 2 present the single-task and multitask results, respectively. Each configuration consists of 10 rollouts across 3 object-instance pairs, totaling 30 evaluations. For POINT BRIDGE, we use 128 points per object extracted using the VLM filtering pipeline . Our key findings are summarized below.

**POINT BRIDGE enables zero-shot sim-to-real transfer with minimal visual alignment.**    As illustrated in Figure 1, our simulation and real-world setups differ significantly in table appearance,

Table 1: POINT BRIDGE enables zero-shot sim-to-real transfer in **single task** settings and shows further performance improvements when trained with small amounts of real-world data.

| Observation Modality | Data Configuration | Bowl on plate | Mug on plate | Stack bowls |
|---|---|---|---|---|
| Image | Real | 9/30 | 10/30 | 11/30 |
| | Co-Train Sim | 2/30 | 17/30 | 14/30 |
| POINT BRIDGE | Real | 25/30 | 25/30 | 24/30 |
| | Zero-Shot Sim | 23/30 | 21/30 | 24/30 |
| | Co-Train Sim | **29/30** | **30/30** | **29/30** |

Table 2: POINT BRIDGE supports both zero-shot sim-to-real transfer and sim-real co-training in multi-task settings. Notably, multi-task learning shows improvements in performance over single-task training.

| Observation Modality | Data Configuration | Bowl on plate | Mug on plate | Stack bowls |
|---|---|---|---|---|
| Image | Real | 10/30 | 11/30 | 11/30 |
| | Co-Train Sim | 6/30 | 10/30 | 15/30 |
| POINT BRIDGE | Real | 22/30 | 26/30 | 24/30 |
| | Zero-shot Sim | 25/30 | 23/30 | 24/30 |
| | Co-Train Sim | **30/30** | **30/30** | **30/30** |

backgrounds, and lighting. Despite these differences, POINT BRIDGE's scene-filtering strategy produces domain-invariant representations, outperforming the strongest baseline by 39% in single-task transfer and 44% in multitask transfer. This stands in contrast to prior approaches, which often require carefully aligned scenes and reality (Maddukuri et al., 2025) or photorealistic simulators (Mittal et al., 2023) to achieve policy transfer. Image-based sim-to-real policies fail entirely in the zero-shot setting, and thus are excluded from the reported results for clarity.

**POINT BRIDGE enables zero-shot sim-to-real policy transfer across diverse object instances.** Figure 1 compares objects used in simulation versus deployment. Even under large discrepancies in visual appearance, POINT BRIDGE requires only minimal object alignment to transfer policies effectively. Additionally, by leveraging FoundationStereo for depth estimation, POINT BRIDGE is able to handle visually challenging objects such as transparent or reflective items, unlike depth sensing from RGB-D cameras, which typically struggles with such items.

**POINT BRIDGE enables multitask zero-shot sim-to-real transfer.** We evaluate both single-task and multitask variants of POINT BRIDGE, where the multitask policy is conditioned on natural language instructions. Since POINT BRIDGE operates on filtered point cloud representations and is language-conditioned, it generalizes naturally to the multitask setting. Empirically, the multitask policy achieves comparable or better performance than single-task policies, demonstrating scalability across diverse tasks.

## 5.3 COMPATIBILITY OF POINT BRIDGE WITH REAL DATA

In this section, we study the effect of jointly training policies with simulated and real-world data. This paradigm, often called *co-training*, has been widely explored in sim-to-real (Maddukuri et al., 2025) and human-to-robot transfer (Haldar & Pinto, 2025). Our key findings are summarized below.

**Co-training with real robot data further improves real-world performance.** We collect 45 teleoperated demonstrations on a real robot for three tasks and jointly train POINT BRIDGE with 1200 simulated demonstrations per task, using an 80–20 simulation-to-real ratio. Results across single-task (Table 1) and multitask (Table 2) show that adding real data consistently boosts performance by up to 30%. By comparison, image-based co-training methods yield a mixed outcome – likely because our simulation and real setups are not as visually aligned as in prior works that assume access to digital-cousin environments in simulation (Maddukuri et al., 2025). Overall, POINT BRIDGE outperforms image-based co-training by 61% in single-task and 66% in multitask settings, highlighting its ability to leverage small amounts of real data alongside large-scale simulation.

Table 4: Study of key designs decisions in POINT BRIDGE.

| Category | Variant | Bowl on plate | Mug on plate | Stack bowls |
|---|---|---|---|---|
| Depth Sensing | Point Tracking | 5/30 | 7/30 | 6/30 |
| | RGB-D | 15/30 | 12/30 | 13/30 |
| | Foundation Stereo | 23/30 | 21/30 | 24/30 |
| Camera alignment | Aligned | 23/30 | 21/30 | 24/30 |
| | Ground truth | 12/30 | 7/30 | 6/30 |

Table 3: Performance of POINT BRIDGE on real tasks with soft and articulated objects.

| Task | Success rate |
|---|---|
| Fold towel | 17/20 |
| Close drawer | 18/20 |
| Bowl in oven | 16/20 |

**POINT BRIDGE supports tasks involving soft and articulated objects.** Table 3 reports results for training single-task POINT BRIDGE policies on three tasks involving soft objects (towel) and articulated objects (drawer, oven). For each task, we collect 20 demonstrations via real robot teleoperation. Overall, POINT BRIDGE achieves an 85% success rate across these tasks, highlighting its effectiveness beyond rigid-object manipulation.

### 5.4 SYSTEM ANALYSIS

Table 4 presents a study of key design decisions in POINT BRIDGE, with insights summarized below.

**Depth estimation for policy inference** During inference, 2D keypoints from the VLM pipeline are lifted to 3D using the depth strategies in Section 4.5. We observe that Foundation Stereo offers the best performance, running at 5 Hz and remaining robust on reflective surfaces. In contrast, multi-view triangulation with MAST3R yields noisy correspondences in dense point clouds, while added point tracking further slows inference to 2.5 Hz with 128 points per object. RGB-D cameras also run at 5 Hz but suffer from noise, missing regions, degraded accuracy at distance, and poor handling of reflective objects. Overall, accurate depth estimation is critical, with stereo vision proving the most reliable and practical choice for sim-to-real transfer in POINT BRIDGE.

**Effect on camera view on policy performance** In simulation, we can uniformly sample ground-truth object points over the entire object, whereas in the real world, point clouds depend on the camera viewpoint and only capture visible surfaces. This creates a mismatch between uniformly sampled points in simulation and view-dependent points in reality. We find that training with camera-aligned points in simulation – generated using real-world camera extrinsics – significantly improves sim-to-real transfer over training on uniformly sampled points.

Additional experiments and analysis have been included in Appendix A.2.2.

### 6 LIMITATIONS & CONCLUSION

In this work, we introduced POINT BRIDGE, a framework that employs domain-agnostic point-based representations to exploit synthetic simulation datasets, enabling zero-shot sim-to-real transfer with minimal visual alignment, supporting co-training with real data, and facilitating multitask policy learning. We recognize a few limitations of this work.

**Limitations** (1) POINT BRIDGE depends on VLMs and other vision models, making it vulnerable to their failures; as these models advance, we expect corresponding improvements in robustness. (2) POINT BRIDGE requires camera pose alignment between simulation and reality to avoid distribution mismatch. A remedy is to train with diverse simulated viewpoints, which can be scaled via synthetic generation tools such as MimicGen (Mandlekar et al., 2023). (3) Point-based abstractions aid generalization but discard critical scene context, limiting performance in cluttered environments. Hybrid representations that preserve sparse contextual cues could address this gap.

## 7 REPRODUCIBILITY STATEMENT

For reproducibility, we have included our experiment hyperparameters along with our hardware specifications and policy through in Appendix A.2. All of our datasets, environments, and training and evaluation code will also be made publicly available.

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

# A APPENDIX

## A.1 COMPARISON WITH POINT POLICY

In this section, we provide a comparison between Point Policy (Haldar & Pinto, 2025) and POINT BRIDGE. While both methods attempt to solve cross-domain policy learning using key points, there are significant differences between the two approaches.

1. While Point Policy primarily focuses on zero-shot human-to-robot transfer, POINT BRIDGE is mainly focused on enabling sim-to-real transfer.

2. Point Policy requires manual human annotations for each task, which limits its scalability. In contrast, POINT BRIDGE leverages a VLM-based pipeline for automated point extraction, enabling it to scale to novel tasks without additional human effort.

3. Point Policy relies on point tracking combined with multi-view triangulation to obtain 3D keypoints. This approach faces two challenges: (1) tracking speed decreases as the number of points increases, and (2) errors in multi-view correspondence can degrade triangulation accuracy. By contrast, POINT BRIDGE employs 2D segmentation tracking with SAM-2 (Ravi et al., 2024), which is fast (20Hz on $512 \times 512$ images) and whose throughput is unaffected by the number of object points. Further, SAM-2 includes a memory module which aids in dealing with occlusions during deployment.

4. In terms of architecture, Point Policy encodes each point track history as an individual transformer token. Instead, POINT BRIDGE uses the PointNet (Qi et al., 2017) encoder to represent the entire 3D point cloud as a single embedding. This design parallels the distinction between ViTs and CNNs for image encoding, where ViTs treat individual pixels as tokens and are generally more data-hungry.

5. While Point Policy is limited to single-task training, POINT BRIDGE functions in multi-task settings.

## A.2 EXPERIMENTS

**Data Generation and Scaling in Simulation** For our simulated experiments, we use the MimicLabs task suite (Saxena et al., 2025) to design 3 atomic tasks, each including 4 different object instance pairs. For every pair, a human demonstrator provides 5 demonstrations, which are scaled up to 300 using MimicGen (Mandlekar et al., 2023). POINT BRIDGE unlocks the potential of such large-scale synthetic data generation by enabling zero-shot sim-to-real transfer. For task design in simulation, we utilize assets from RoboCasa (Nasiriany et al., 2024), focusing primarily on pick-and-place task transfer from simulation. More complex articulated tasks (e.g., opening ovens) are difficult to transfer to the real world due to unrealistic asset dynamics – for instance, simulated ovens often open with a simple handle push unlike real ovens that require pressing a button at varying locations. Addressing this gap would require more realistic simulation assets (Lightwheel, 2025) along with training across diverse object variants, which we leave for future work. In this work, we primarily focus on establishing visual invariance across simulation and the real world, enabling cross-domain zero-shot policy transfer.

**Considerations for Policy Learning** Our experiments use ZED 2i stereo cameras with depth estimated via FoundationStereo. While the vanilla model for FoundationStereo is slow for closed-loop control, the TensorRT-optimized version achieves up to 10 Hz on an NVIDIA RTX 5090 GPU. Since this GPU resides on a separate machine from the robot, we use high-speed Ethernet for low-latency communication, primarily for image transfer, resulting in an overall control frequency of 5 Hz. When using depth from an RGB-D camera, the models run directly on the robot's NVIDIA Quadro RTX 8000 GPU, also operating at 5 Hz.

**Task Descriptions** We evaluate POINT BRIDGE across a diverse set of tasks, with rollouts on the real robot depicted in Figure 3. Each task involves substantial spatial variation and multiple distinct object instances, with significant differences between the simulation and real objects. For the tasks `bowl on plate`, `mug on plate`, and `stack bowls`, we generate 1200 demonstrations in simulation spanning four object-instance pairs. For co-training, we supplement this with 45

teleoperated demonstrations in the real world across three additional object pairs, illustrating cross-domain variability. For real tasks such as `fold towel`, `close drawer`, and `put bowl in oven`, we only collect real-world data (20 demonstrations on a real robot), as aligned simulation assets are unavailable.

### A.2.1 HYPERPARAMETERS

The hyperparameters for POINT BRIDGE have been provided in Table 5.

Table 5: List of hyperparameters.

| Parameter | Value |
| --- | --- |
| Learning rate | $1e^{-4}$ |
| Image size | $672 \times 448$ (for Foundation Stereo Tensor RT version) |
| Batch size | 16 |
| Optimizer | Adam |
| Number of training steps | 300000 |
| Hidden dim | 256 |
| Observation history length | 1 |
| Action head | Deterministic |
| Action chunk length | 40 (with training data at 10Hz) |
| # keypoints per object | 128 |

### A.2.2 ADDITIONAL EXPERIMENTS AND SYSTEM ANALYSIS

**Comparison with point cloud and point track baselines**   We compare the single-task, zero-shot sim-to-real transfer performance of POINT BRIDGE with a point cloud baseline, BAKU-PCD, and a point track baseline, Point Policy (Haldar & Pinto, 2025). For BAKU-PCD, we use the BAKU (Haldar et al., 2024) architecture with unfiltered point cloud inputs containing 512 scene points, encoded using a PointNet (Qi et al., 2017) encoder similar to POINT BRIDGE. We observe that including the table and the curtains surrounding the real robot setup results in zero success rates, so we manually restrict the work area to exclude these elements from the point cloud for BAKU-PCD. Notably, this kind of filtering is performed automatically by the VLM-based point extraction pipeline in POINT BRIDGE. Point Policy (Haldar & Pinto, 2025) uses a sparse set of semantically meaningful points, labeled by a human user on a canonical image, as input. At evaluation time, it uses semantic correspondence to locate the corresponding points in the target scene, and then Co-Tracker (Karaev et al., 2023) is used to track these initialized points across the trajectory. We find that semantic correspondence between simulated and real images performs poorly, resulting in zero success rates for Point Policy in the zero-shot sim-to-real setting. These results have been presented in Table 6. Overall, we find that POINT BRIDGE's automated keypoint extraction enables significantly more robust sim-to-real transfer than previous point cloud and point track baselines.

**Sensitivity to calibration changes between training and deployment**   The results in Table 1 and Table 2 assume that the camera viewpoints are identical between simulation and the real world. To relax this assumption, we synthetically generate the segmented point clouds used by POINT BRIDGE from eight distinct camera views positioned around the robot in simulation. For each view, we extract a 3D segmented point cloud and transform it to the robot's base frame using the relevant camera extrinsics. This variation captures different occlusion patterns and increases the model's robustness to camera viewpoint changes that may occur between training and deployment. The results in Table 7 evaluate single-task zero-shot sim-to-real transfer under such viewpoint variations between simulation and real. Notably, even without matched camera viewpoints between simulation

Table 6: Comparison between single-task zero-shot sim-to-real performance of POINT BRIDGE and baselines using an unfiltered point cloud and point tracks as input.

| Method | Bowl on plate | Mug on plate | Stack bowls |
|---|---|---|---|
| BAKU-PCD | 6/30 | 9/30 | 12/30 |
| Point Policy (Haldar & Pinto, 2025) | 0/30 | 0/30 | 0/30 |
| POINT BRIDGE | **23/30** | **21/30** | **24/30** |

Table 7: Comparison between single-task zero-shot sim-to-real performance of POINT BRIDGE with and without identical camera views between simulation and the real-world.

| POINT BRIDGE | Bowl on plate | Mug on plate | Stack bowls |
|---|---|---|---|
| w/ identical camera views | **23/30** | **21/30** | **24/30** |
| w/ randomized camera views | 12/30 | 12/30 | 18/30 |

and reality, POINT BRIDGE attains an average success rate of approximately 47% across three tasks. We observe a drop in performance when transitioning from matched to randomized viewpoints. This opens up an important direction for future work: developing methods that achieve robustness to viewpoint-dependent discrepancies in 3D point distributions for policy learning.

**Effect of background distractors**    To evaluate the robustness of our scene filtering pipeline (Section 4.3), we compare zero-shot single-task sim-to-real transfer performance for BAKU-PCD (using unfiltered point cloud inputs; see "Comparison with point cloud and point track baselines" earlier) and POINT BRIDGE both with and without background distractors. Results are presented in Table 8, with representative distractor examples shown in Figure 4. We observe that BAKU-PCD, relying on unfiltered point clouds, is highly susceptible to distractor objects, yielding a zero success rate under these conditions. In contrast, POINT BRIDGE, which incorporates scene filtering, maintains performance on par with the distractor-free scenario and exhibits strong robustness to background clutter.

**Generalization to held-out objects**    As shown in Table 1 and Table 2, POINT BRIDGE demonstrates strong zero-shot sim-to-real transfer to novel real-world objects unseen in simulation, achieving 76% and 80% success rates in single-task and multi-task scenarios, respectively. Co-training with both simulated and real data further raises success rates to 98% (single task) and 100% (multi-task) for objects present in the real dataset, likely due to reducing geometric disparities between synthetic and real instances. Beyond these results, we also evaluate the co-trained POINT BRIDGE policies on held-out objects missing from both simulated and real training sets – a stricter measure of generalization to novel object instances. As shown in Table 9, multi-task success rates on held-out objects remain high at 97% on unseen objects compared to 100% for those encountered during training, with failures mainly occurring for bowls which were much larger than those in the training data. These results highlight the robustness of POINT BRIDGE to entirely novel object instances. Rollout videos on held-out object instances are available on https://pointbridge-anon.github.io/.

**Effect of number of points**    We evaluate the impact of the number of object points on policy performance in simulation across three configurations: 10, 64, and 128 points per object. These results are included in Table 10. While overall performance remains similar across these settings, 64 points per object yields the best performance. Notably, all configurations achieve over 86% success, demonstrating that POINT BRIDGE is effective across both sparse and dense point cloud regimes.

**Effect of action representation**    We compare the performance of POINT BRIDGE across two action representations: pose regression and point track prediction. For point track prediction, we follow Point Policy (Haldar & Pinto, 2025) and predict a future chunk of end-effector points (described in Section 4.3) instead of future pose sequences. These results are reported in Table 10. Unlike Point Policy, which reported gains from point track prediction, we observe comparable performance between the two representations. A likely reason is the difference in dataset scale – Point Policy used

Table 8: Comparison between single-task zero-shot sim-to-real performance of POINT BRIDGE and BAKU-PCD in the presence of background distractors.

| Method | Background distractors | Bowl on plate | Mug on plate | Stack bowls |
|--------|:----------------------:|:-------------:|:------------:|:-----------:|
| BAKU-PCD | ✗ | 6/30 | 9/30 | 12/30 |
|  | ✓ | 0/30 | 0/30 | 0/30 |
| POINT BRIDGE | ✗ | 23/30 | 21/30 | 24/30 |
|  | ✓ | 22/30 | 20/30 | 25/30 |

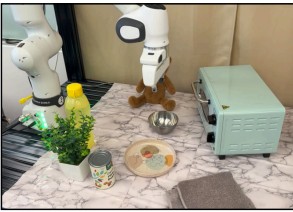 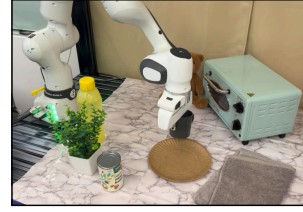 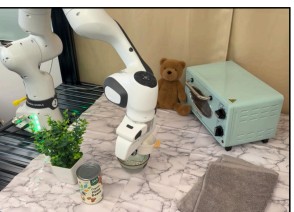

Bowl on plate          Mug on plate          Stack bowls

Figure 4: Examples of background distractors in real-robot setup.

at most 30 demonstrations per task, whereas our experiments leverage 1200 simulated demonstrations per task, potentially reducing the relative benefit of point track supervision.

**Latency analysis for VLM-guided scene filtering pipeline**    Table 11 summarizes the measured runtimes for each component of the VLM-guided scene filtering pipeline. The analysis is divided into two phases: initialization and per-step execution. During initialization, all required models are loaded and executed at the start of the trajectory, taking approximately 9 seconds – a one-time overhead that occurs only before the first policy step and is thus acceptable. For subsequent steps, only SAM2 (Ravi et al., 2024) (object mask tracking) and Foundation Stereo (Wen et al., 2025) (depth computation) are invoked, bringing the per-step runtime down to around 0.115 seconds and enabling real-time policy deployment.

**Robustness analysis for VLM-guided scene filtering pipeline**    For all tasks in this work, the VLM-guided scene filtering pipeline consistently achieves high success rates. We do not filter our reported results for failures of the used VLMs or vision foundation models (VFMs). Hence, all success rate obtained are despite any foundation model failures that might occur. To quantify the robustness, we consider the three sim-to-real tasks - *bowl on plate*, *mug on plate*, and *stack bowls* - and place the objects in 20 randomized positions. For each position, we deploy the scene extraction pipeline and record filtering successes and failures. For *bowl on plate*, there was only one failure among the 20 trials when the metallic bowl was occluded by the robot gripper in its initiazation position. For *mug on plate*, all trials succeeded in filtering out the mug and the plate on the table. For *stack bowls*, there was only one failure among the 20 trials where Molmo Deitke et al. (2024) could not find the small white bowl placed on the table. These failure cases have been illustrated in Figure 5. Despite very low VLM error rates for the tasks considered in the paper, we acknowledge that the failure rates might go up, especially with cluttered scenes or scenes with multiple similar-looking objects. A systematic study of VLM failures would be interesting for future research.

Table 9: Performance of multi-task co-trained POINT BRIDGE on a held-out set of object instances introduced at test time.

| POINT BRIDGE | Bowl on plate | Mug on plate | Stack bowls |
|---|---|---|---|
| Same objects | 30/30 | 30/30 | 30/30 |
| Held-out objects | 28/30 | 29/30 | 30/30 |

Table 10: A systematic analysis of the effect of the number of object points and action representation on POINT BRIDGE performance

| Category | Variant | Bowl on plate | Mug on plate | Stack bowls |
|---|---|---|---|---|
| # Object points | 10 | 0.95 | 0.8 | 0.92 |
| | 64 | 0.96 | 0.95 | 0.92 |
| | 128 | 0.9 | 0.8 | 0.9 |
| Action prediction | Pose | 23/30 | 21/30 | 24/30 |
| | Points | 24/30 | 24/30 | 24/30 |

Table 11: Latency analysis of the VLM-guided scene filtering pipeline

| Mode | Step | Time (in seconds) |
|---|---|---|
| Initialization | Gemini Query | ~1.95 |
| | Molmo | ~4.8 |
| | SAM Init | ~2.4 |
| | **Total time** | **~9.15** |
| Per step | Foundation Stereo | ~0.07 |
| | SAM Tracking | ~0.045 |
| | **Total time** | **~0.115** |

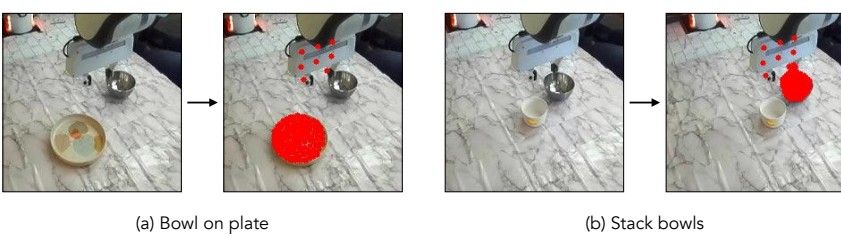

(a) Bowl on plate          (b) Stack bowls

Figure 5: Examples of failure cases of the VLM-guided scene filtering pipeline.

