# OpenReview forum: "Point Bridge: 3D Representations for Cross Domain Policy Learning"
_ICLR.cc/2026/Conference — Submitted to ICLR 2026_

### Official Review · Reviewer_h3iK · 2025-10-27

**Soundness:** 3
**Presentation:** 4
**Contribution:** 3
**Rating:** 6
**Confidence:** 4

**Summary:**

The paper proposes **Point Bridge**, a framework that bypasses the visual domain gap by learning policies on a unified, domain-agnostic 3D point-based representation. Instead of training on raw pixels, the policy operates on 3D point clouds representing the task-relevant objects and the robot's end-effector.
The method achieves zero-shot sim-to-real transfer and significantly improves the task success rate over the baseline using raw pixel inputs. They also show that the performance could be further boosted by co-training with a small amount of real-world data.

**Strengths:**

1. **Clear, Well-motivated Idea**: Using compact 3D keypoint representations to move the sim/real gap from image appearance to a geometric representation is intuitive and practically appealing. The idea connects naturally to data-generation systems like MimicGen that scale up simulated demonstrations.

2. **Strong Empirical Results**: Extensive real-robot rollouts, zero-shot sim-to-real experiments, and sim+real co-training analyses with multiple ablations give evidence the approach improves success rates compared to an image-based baseline.

3. **Clarity**: The paper is exceptionally well-written and easy to follow.

**Weaknesses:**

1. **Dependence on Foundation Models**: The perception pipeline is composed of multiple foundation models (Gemini, Molmo, SAM2, and Foundation Stereo) without a verification mechanism. The performance may be affected if any model fails.

2. **The Domain Gap is Shifted, Not Removed**: By discarding RGB and using only geometry, the method reduces appearance mismatch but increases reliance on accurate depth and occlusion handling. Physics and contact dynamics gaps between sim and real remain, so claims about “closing the sim-to-real gap” should be tempered.

3. **Limited Task Scope**: The experiment tasks are mostly pick-and-place tasks without complex dynamic, geometry, and environment constraints. Although the paper includes "Put bowl in oven" as the articulation task, the robot does not seem to learn the articulation to close the oven door in Figure 3 and the video.

4. **Weak Baseline Comparison**: The image-based baseline is trained on a dataset generated from MimicLab, whose visual rendering gap is notably bigger than the current state-of-the-art simulators (e.g., Issac Sim). The baseline model does not fully demonstrate the limit of image-based sim-to-real transfer. There is also a missing direct empirical comparison to other keypoint/point-based representations or the closest prior works that use learned keypoints or structured geometric inputs.

5. **No Failure Analysis**: Results are reported as success counts without a structured failure taxonomy, which helps understand robustness and reproducibility.

**Questions:**

1. The policy uses only 3D keypoint positions without RGB. Does this create ambiguity for more complex tasks? How does this design handle tasks that require appearance cues (e.g., sorting by color or distinguishing visually similar objects with the same geometry)? Prior work [1] indicates that combining visual features with geometry can help. Could you elaborate on this?

2. What are the independent failure modes and success rates of the perception pipeline? How do compounded errors (e.g., Gemini failing to identify the correct object, SAM2 failing to segment, or FoundationStereo producing noisy depth) propagate to the policy? Is it possible to develop a verification mechanism to make it more robust?

3. Does the current perception pipeline really capture "task-relevant" keypoints? The current method seems to be simply sampling all points on the task object without understanding the task. For instance, in the task "closing the drawer", the points on the handle and the corners of the drawer should be enough to solve the task and infer the articulation. Would a small set of semantically relevant keypoints perform as well or better? How does the method generalize to variations in object geometry (wider/taller/deeper drawers) without infinitely creating more assets in the simulation?

4. Since the model uses geometry only, what specific benefits does a small amount of real data provide? Is the improvement due primarily to the exact same geometry of the object?

**Typos**:
1. In Table 2’s caption, should “single task” be “multitask” (line 373)?
2. The order of Table 3 and Table 4 appears inverted.

[1] Fang, Xiaolin, et al. "KALM: Keypoint Abstraction Using Large Models for Object-Relative Imitation Learning." 2025 IEEE International Conference on Robotics and Automation (ICRA). IEEE, 2025.

---

> ### Author Response · Authors · 2025-11-21
> **Rebuttal**
>
> We thank the reviewer for the detailed comments about the work. We are glad that you found the idea clear and well-motivated, the results strong, and the writing clear. We would like to provide clarifications to the questions below:
>
> **Dependence on Foundation Models:** Please see point (F) in the global response.
>
> **The Domain Gap is Shifted, Not Removed:** In this work, we primarily address the visual domain gap between simulation and the real world, and assume that the dynamics gap is small. We acknowledge this point and have replaced “minimal alignment” with “minimal visual alignment” in the paper.
>
> **Limited Task Scope:** Please see point (G) in the global response.
>
> **Weak Baseline Comparison:** Please see point (C) in the global response. We thank the reviewer for bringing this up since the updated results now demonstrate the effectiveness of Point Bridge against a more comprehensive set of baselines.
>
> **No Failure Analysis:** Please see point (E) in the global response regarding concerns with respect to VLM failures. Regarding policy failures, failure cases typically include grasping failures, reaching incorrect positions, failure to place in the correct location, and failure to fully close the drawer (in case of drawer closing).
>
> We hope we have been able to address your questions in the above clarifications. Kindly let us know if you have additional questions or concerns. If not, please consider increasing your rating.

---

### Official Review · Reviewer_7sFb · 2025-10-31

**Soundness:** 3
**Presentation:** 3
**Contribution:** 2
**Rating:** 4
**Confidence:** 3

**Summary:**

This paper introduces Point Bridge, a novel framework designed to address the sim-to-real transfer bottleneck in robotic manipulation. The core innovation is the use of a unified, domain-agnostic point-based representation to bridge the visual domain gap. By leveraging VLMs for automated extraction of task-relevant 3D keypoints, Point Bridge distills observations into a compact point cloud, upon which transformer-based policies are trained. The paper demonstrates that this approach enables effective zero-shot sim-to-real transfer using purely synthetic data from tools like MimicGen. Furthermore, it shows that performance can be substantially enhanced through co-training with small amounts of real-world data and that the framework naturally facilitates multitask learning. Extensive real-world experiments on several manipulation tasks report significant improvements over prior methods.

**Strengths:**

1.The central idea of a point-based representation is both powerful and elegant. It directly attacks the problem of visual domain gap by moving away from pixel-level inputs to a more geometric abstraction. This is a more scalable approach than striving for photorealistic simulation.

2.The framework presents a complete and highly automated pipeline. It intelligently integrates synthetic data generation, VLM-guided scene filtering, and modern policy learning into a cohesive system. The automation of point extraction, removing the need for manual annotation, is a particularly crucial contribution for practical adoption.

3.The experimental validation is thorough and compelling. The evaluation covers not only zero-shot transfer but also co-training and multitask scenarios, showcasing the flexibility of POINT BRIDGE. The large number of real-world evaluations lends significant credibility to the results. The ablation studies on depth estimation strategies and the importance of camera-aligned point sampling in simulation provide valuable insights into the system’s engineering nuances.

**Weaknesses:**

1.POINT BRIDGE exhibits a strong dependence on external pre-trained vision models. The entire pipeline’s entry point relies on models like Gemini and SAM2. Consequently, the robustness of POINT BRIDGE is inherently tied to the performance of these components, and any failures in perception cannot be easily corrected within the framework itself.

2.The framework relies on assumptions about a calibrated scene with known camera intrinsics and extrinsics. This requirement for a consistent reference frame might limit deployment in more dynamic setups where camera poses are not fixed or precisely known.

3.The abstraction into point clouds, while beneficial for generalization, can lead to a loss of critical scene context. The paper itself notes that this can limit performance in cluttered environments, as fine-grained visual details or contextual cues necessary for disambiguation might be discarded.

**Questions:**

1.Could you provide more detail on the failure cases and robustness of the VLM-guided pipeline? For instance, how often did the initial object identification or the subsequent segmentation with SAM2 fail or produce inaccurate results in your real-world trials? A discussion of common failure modes and whether the system has any inherent mechanisms to detect or mitigate them would be very helpful.

2.The choice of 128 points per object is noted. Was this parameter systematically ablated? It would be interesting to know if there is a point of diminishing returns or if certain tasks benefit from a different number of points. Furthermore, was any sampling strategy beyond uniform sampling explored that might better capture object geometry for dexterous manipulation?

3.The paper repeatedly emphasizes minimal visual and object-level alignment. To better understand the boundaries of this claim, could you clarify what minimal object alignment entails? Does it allow for transferring policies between objects of different categories with entirely different geometries, or does it assume functional and rough geometric similarity? Showcasing results on a task with extreme object shape variation between sim and real would powerfully reinforce this point.

---

> ### Author Response · Authors · 2025-11-21
> **Rebuttal**
>
> We thank the reviewer for the detailed comments about the work. We are glad that you found the point-based representations to be elegant, the pipeline to be complete, and the experimental results to be thorough and compelling. We provide clarifications to the questions below:
>
> **“... strong dependence on external pre-trained vision models”:** Please see point (F) in the global response.
>
> **“... relies on assumptions about a calibrated scene … limit deployment in more dynamic setups …”:** Please see point (D) in the global response for concerns regarding changes in calibration between training and deployment. In scenarios involving dynamic camera poses where the cameras are mounted on a robot’s end effector, the relative pose of the camera with respect to the end effector is typically known from the 3D mount design (and if not, a calibration procedure can be used to estimate it). Consequently, the camera pose relative to the robot base can be derived at any time as the robot moves in 3D space. For cases where the camera is external and moves independently of the robot base, we can either use tracking cameras such as the Intel RealSense T265 or employ SLAM or other learning-based methods to continuously estimate camera poses. However, the accuracy and reliability of such tracking solutions require further evaluation before deployment in dynamic environments.
>
> **“... loss of critical scene context”:** As described in Section 6, scene filtering results in loss of scene context, which can hurt performance in cluttered scenes. A potential solution for this is to design hybrid representations that include important objects in the scene while including obstacles that the robot needs to avoid during deployment.
>
> **“... failure cases and robustness of the VLM-guided pipeline”:** Please see point (E) in the global response. We thank the reviewer for raising this concern since we now have more complete results on the framework throughput and robustness.
>
> **Number of object points and sampling strategy:** We have already ablated the effect of the number of object points on policy performance in Appendix A.2.2 (Table 10). We observe that for tasks in this work, Point Bridge is effective across both sparse and dense point cloud regimes. Regarding sampling strategies, we have only experimented with uniform sampling. More clever sampling schemes, such as semantic clustering [1], can be explored in future work.
>
> **Meaning of minimal object alignment:** For each task in this paper, we train on several object variants differing in shape and size, and evaluate on objects that are similar – but not identical – to those used in training. In the new results presented in Appendix A.2.2 (Table 9), we further evaluate the co-trained policies on a held-out set that includes objects with substantial differences from the training set, such as much larger bowls. Videos of these rollouts are available on our webpage.
>
> We hope we have been able to address your questions in the above clarifications. Kindly let us know if you have additional questions or concerns. If not, please consider increasing your rating.
>
> **References**
>
> [1] Wang, Shengjie, et al. "SKIL: Semantic keypoint imitation learning for generalizable data-efficient manipulation." arXiv preprint arXiv:2501.14400 (2025).

---

### Official Review · Reviewer_g2VG · 2025-10-31

**Soundness:** 3
**Presentation:** 3
**Contribution:** 1
**Rating:** 0
**Confidence:** 4

**Summary:**

The manuscript Point Bridge proposes to use a 3D point cloud representation as the basis of a robotic policy. By utilizing this approach the authors are able to bride training on simulation data with deployment on real robots. The advantages of simulation data are utilized by the use of MimicGen, increasing the size of the training dataset substantially. On real data, object- and robot-centric point clouds are generated using multi-view stereo and SAM2 for object segmentation. Performance is evaluated in simulation and on real data by comparing to an image-only baseline, showing substantial improvements on the same amount of training data.

**Strengths:**

- The method shows good transfer from simulation training to real-world deployment.
- The proposed pipeline is well-engineered, utilizing powerful open-vocabulary models likely capable of generalization to broader scenarios.

**Weaknesses:**

- The work has very limited novelty. Point cloud and point track representations have been used in numerous previous works (as cited by the authors), for specialist policies as well as of generalist VLA models. While these works do not explicitly target the sim2real problem, they show capable policies on simulation data, human demonstrations and real robot demonstrations. In this context, especially human demonstration data also represents a domain transfer problem.
- The work does not compare to any point cloud and point track-based method. Such a comparison could demonstrate the potential advantage of the proposed pipeline over the baseline methods on the sim2real problem.
- The method requires a calibrated depth/stereo camera setup, with no change between training and inference. The authors do not evaluate the sensitivity of calibration change between training and deployment.
- Minor: Tables 1/2 have incorrect captions, with both being labeled as single task.

**Questions:**

- A list of differences w.r.t. the most important point tracking works would help the reader to position the work's contributions.
- A benchmark against these methods in the domain transfer case would show the advantages of the proposed method.
- It is unclear how strongly the method depends on a match of camera calibration between training and inference.

---

> ### Author Response · Authors · 2025-11-21
> **Rebuttal**
>
> We thank the reviewer for the detailed comments about the work. We are glad that you found the approach well engineered and showcasing good transfer from sim to real. We would like to provide clarifications to the questions below:
>
> **“...  limited novelty”:** Please see point (B) in the global response.
>
> **“... does not compare to any point cloud and point track-based method”:** Please see point (C) in the global response. We thank the reviewer for bringing this up since the updated results now demonstrate the effectiveness of Point Bridge against a more comprehensive set of baselines.
>
> **“... do not evaluate the sensitivity of calibration …”:** Please see point (D) in the global response. We appreciate the reviewer’s feedback, which led us to develop and test a relaxed variant of Point Bridge. This version demonstrates zero-shot sim-to-real transfer even when camera viewpoints differ between training and deployment.
>
> **Incorrect Captions:** Thank you for bringing this to our notice. We have corrected the captions for Table 1 and Table 2 in the updated version of the paper.
>
> **Difference with existing point tracking work:** We have already included a comparison with Point Policy, the most relevant point track baseline, in Appendix A.1.
>
> We hope we have been able to address your questions in the above clarifications. Kindly let us know if you have additional questions or concerns. If not, please consider increasing your rating.

---

### Official Review · Reviewer_oEVA · 2025-11-01

**Soundness:** 3
**Presentation:** 3
**Contribution:** 3
**Rating:** 4
**Confidence:** 4

**Summary:**

This paper proposes POINT BRIDGE, a cross-domain policy learning framework that centers on “task-relevant points” (keypoints / sparse point clouds) as a unified 3D representation for sim-to-real transfer. The system uses a VLM-guided pipeline (segmentation + depth/reconstruction) to extract task-relevant 3D points (via FoundationStereo / RGB-D / stereo triangulation / tracking), then feeds them to a PointNet + Transformer (BAKU) policy. The goal is zero-shot sim-to-real from large-scale synthetic training, with further gains from limited real co-training.

**Strengths:**

1.Unified point representation: Mapping both sim and real to task-relevant points is pragmatic and deployment-friendly.

2.Empirical gains: The approach improves over image-based baselines in both zero-shot and limited co-train regimes.

3.Systematic ablations: The paper compares multiple depth/reconstruction sources and discusses viewpoint alignment, offering evidence for deployment trade-offs (success vs. frequency).

4,Implementation clarity: The synthetic-to-3D-to-policy pipeline is clearly described and appears reproducible.

**Weaknesses:**

1.Limited novelty: Most components (data generation, object filtering, depth estimation, policy learning) are existing modules strung together; the main contribution is a well-engineered integration and representation choice rather than a new learning principle.

2.Baseline coverage (3D/depth): Comparisons are primarily against image-based policies. Missing are baselines that take dense point clouds/depth directly (e.g., point-cloud based, depth-only Diffusion/BC variants) under matched data—making it hard to claim that POINT BRIDGE universally outperform other input modalities.

3.Task simplicity: Core evaluations focus on pick-and-place / stacking, leaving uncertainty about performance on high-contact, non-rigid, assembly, or constrained tasks. I would like to see more results on more complex tasks.

4.Latency/robustness under-analyzed: FoundationStereo yields ~5 Hz, additionally the paper applys many foundation models, The latency and robustness of the pipeline should be more seriously analyzed.

**Questions:**

Please see 2.3.4 in weaknesses.

---

> ### Author Response · Authors · 2025-11-21
> **Rebuttal**
>
> We thank the reviewer for the detailed comments about the work. We are glad that you found the approach pragmatic and deployment friendly and the ablations systematic. We would like to provide clarifications to the questions below:
>
> **Limited Novelty:** Please see point (B) in the global response.
>
> **Baseline Coverage:** Please see point (C) in the global response. We thank the reviewer for bringing this up since the updated results now demonstrate the effectiveness of Point Bridge against a more comprehensive set of baselines.
>
> **Task Simplicity:** Please see point (G) in the global response.
>
> **Latency/Robustness underanalyzed:** Please see point (E) in the global response. We thank the reviewer for raising this concern since we now have more complete results on the framework throughput.
>
> We hope we have been able to address your questions in the above clarifications. Kindly let us know if you have additional questions or concerns. If not, please consider increasing your rating.

---

### Author Response · Authors · 2025-11-21
**Global response**

We thank the reviewers for their constructive feedback. We are glad that the reviewers found our approach pragmatic (reviewer oEVA), well engineered (reviewers oEVA, g2VG), and scalable (reviewer 7sFb) with strong empirical results (reviewers oEVA, 7sFB, h3iK). The reviewers have requested additional experiments and important clarifications. **We have also revised the paper with additional experimental results in Appendix A.2.2.** We have provided detailed responses to each review, along with a summary of some shared concerns in this global response.

**(A) New experimental results:** We have added several new experimental results in Appendix A.2.2 to provide clarifications for reviewer questions as well as further establish the robustness of Point Bridge. The new results include: **(a)** Comparison with point cloud and point track baselines (Reviewers oEVA, g2VG, h3iK). **(b)** A study of sensitivity to calibration changes between training and deployment (Reviewers g2VG, 7sFb). **(c)** Effect of background distractors on policy performance. **(d)** A study of generalization to a novel, held-out set of object instances (Reviewer h3iK). **(e)** An analysis of latency and robustness of the VLM pipeline (Reviewers oEVA, 7sFb, h3iK).

**(B) Concerns about novelty (Reviewers oEVA, g2VG):** We acknowledge concerns about the novelty of each individual component in Point Bridge; however, Point Bridge as a whole constitutes a first-of-its-kind turn-key framework for sim-to-real manipulation that offers significant value to the community. We will also be releasing the code for reproducibility. Our work identifies key design choices that enable robust zero-shot sim-to-real transfer, with simple modifications delivering a 44% gain in zero-shot transfer and up to 66% when co-trained with limited real data. The simplicity and modularity of our framework is a core advantage, making it easy to adapt and extend. Unlike prior point cloud or point track methods that require extra manual annotations for scene filtering [1, 2] or rely on large unfiltered datasets [3], Point Bridge automates scene segmentation without annotation overhead. Addressing reviewer g2VG’s comment that “human demonstration data also represents a domain transfer problem”, existing approaches still require manual human annotation [1], graph construction [2], or co-training with robot-collected data [4, 5]. Our framework unifies automated scene filtering, robust point cloud generation in arbitrary scenes, and policy training, resulting in policies that are agnostic to background or distractor changes (see new experiments in Appendix A.2.2).

**(C) Concerns about baseline coverage (Reviewers oEVA, g2VG, h3iK):** We address this concern by providing additional results in Appendix A.2.2. We add an unfiltered point cloud baseline with 512 points per scene for the single-task zero-shot sim-to-real setting and observe that without scene filtering, the policy performance drops significantly. It is worth noting that to make this baseline work, we had to filter the point cloud to remove the table and curtains, without which this would not work at all due to the differences between simulated and real scenes. Notably, this kind of filtering is performed automatically by the VLM-based point extraction pipeline in Point Bridge. We also add a point track baseline, Point Policy [1], which uses sparse human annotations with semantic correspondence for labelling key points on a new scene image. As shown in Appendix A.2.2, semantic matching between simulated and real-world images fails to yield accurate keypoint labels, resulting in poor performance for this baseline.

**(D) Concerns about sensitivity to calibration (Reviewers g2VG, 7sFb):** The results in Table 1 and Table 2 assume that the camera viewpoints are identical between simulation and the real world. To relax this assumption, we provide new results in Appendix A.2.2 where we synthetically generate the segmented point clouds used by Point Bridge from eight distinct camera views positioned around the robot in simulation. For each view, we extract a 3D segmented point cloud and transform it to the robot’s base frame using the relevant camera extrinsics. This variation captures different occlusion patterns and increases the model’s robustness to camera viewpoint changes that may occur between training and deployment. The results in Table 7 evaluate single-task zero-shot sim-to-real transfer under such viewpoint variations between simulation and real. Notably, even without matched camera viewpoints between simulation and reality, Point Bridge attains an average success rate of approximately 47% across three tasks. While we observe a drop in performance when transitioning from matched to randomized viewpoints, this opens up an important opportunity for future work: developing methods that achieve increased robustness to viewpoint-dependent discrepancies in 3D point distributions for policy learning.

---

> ### Author Response · Authors · 2025-11-21
> **Global response (continued)**
>
> **(E) Concerns about latency and robustness of VLM pipeline (Reviewers oEVA, 7sFb, h3iK):** We have included new results in Appendix A.2.2 studying the latency and robustness of the VLM-guided scene filtering pipeline. We thank the reviewers for raising this concern since we now have more complete results on the framework throughput. To summarise these results, we observe that while the VLM pipeline needs around 9 seconds for initialisation (a one-time process for each rollout), it only requires ~0.115s for per-step execution, enabling real-time policy rollouts. Further, we observe that for the tasks studied in the paper, the VLM pipeline exhibits extremely low error rates. For more details, we refer the reader to Appendix A.2.2.
>
> **(F) Dependence on pre-trained vision models (Reviewers 7sFb, h3iK):** While we agree that Point Bridge heavily relies on the performance of VLMs and vision foundation models (VFMs), it is important to note that these models exhibit excellent performance in our settings, achieving close to zero error rates in our experiments (robustness analysis in Appendix A.2.2). We have also tried the same pipeline on a cluttered scene and observed similar extremely low error rates. As these models keep getting better, these error rates will go down even further, and they can be potentially applied to more complicated scenes.
>
>  **(G) Concerns about task simplicity (Reviewers oEVA, h3iK):** We acknowledge that all sim-to-real results presented in the paper are pick-and-place tasks. The primary reason behind this is that the number of variations of articulated objects (for example, ovens) is currently limited in our simulators. Scaling this up with more simulation assets is an interesting direction for future research. For this work, we restrict the sim-to-real experiments to rigid body assets prevalent in both sim and real. Since Point Bridge proposes a unified keypoint-based representation across domains, we are still able to demonstrate applicability to soft and articulated objects through experiments with real data (Section 5.3, Table 3). Consequently, Point Bridge can leverage complementary data distributions across domains.
>
>
> **References**
>
> [1] Haldar, Siddhant, and Lerrel Pinto. "Point policy: Unifying observations and actions with key points for robot manipulation." arXiv preprint arXiv:2502.20391 (2025).
>
> [2] Zhu, Yifeng, et al. "Vision-based manipulation from single human video with open-world object graphs." arXiv preprint arXiv:2405.20321 (2024).
>
> [3] Yang, Rujia, et al. "Fp3: A 3d foundation policy for robotic manipulation." arXiv preprint arXiv:2503.08950 (2025).
>
> [4] Qiu, Ri-Zhao, et al. "Humanoid Policy~ Human Policy." arXiv preprint arXiv:2503.13441 (2025).
>
> [5] Yang, Ruihan, et al. "Egovla: Learning vision-language-action models from egocentric human videos." arXiv preprint arXiv:2507.12440 (2025).

---

### Meta-Review · Area_Chair_2XUD · 2026-01-07

**Summary:**

Overall, this paper is a borderline paper given the reviews and rebuttal. Strengths include great presentation and engineering works, while weaknesses, such as the limited novelty and experiments, are commonly acknowledged by all the reviewers. The author supplements more results and explanations regarding the weaknesses in the rebuttal, which may partly address some of these concerns. However, the task simplicity, especially in the current community where most work even pursues a generalist policy, is still a major weakness for such a modular method. Therefore, I vote for rejection given its current state and recommend supplementing more baselines and experiments to support that a modular method can have a strong performance with great generalizability across different tasks and environments.

**Reviewer Concerns:**

As mentioned above, most of the reviewers mentioned the limited novelty, lack of enough experiments for diverse tasks and settings and comparison with more baselines. Some other concerns include the detailed information like the latency and robustness of the method and the sensitivity to calibration. The rebuttal supplements some convincing results for comparison with other alternative baselines and related explanations. However, the supplemented experiments and baselines are still limited to support the generalizability of the effectiveness of the modular method. The novelty problem is also inherent and may affect the final review scores given different opinions of reviewers.

**Reviewer Scores:**

The initial review scores are 4, 0, 4, 6, among which the score of 0 can be a little biased from the real feedback given the related reviews. After the reviews, the scores can be expected to maintain the overall average level (around 4 or lower), considering that the main concerns regarding novelty and experiments cannot be fully addressed. The final decision also agrees with the majority of opinions among reviewers.

---

### Decision · Program_Chairs · 2026-01-26

Reject